# Systematic Review of Patient-Reported Outcomes following Surgical Treatment of Lymphedema

**DOI:** 10.3390/cancers12030565

**Published:** 2020-02-29

**Authors:** Michelle Coriddi, Joseph Dayan, Nikhil Sobti, David Nash, Johanna Goldberg, Anne Klassen, Andrea Pusic, Babak Mehrara

**Affiliations:** 1Memorial Sloan Kettering Cancer Center, New York, NY 10065, USA; dayanj@mskcc.org (J.D.); goldbej2@mskcc.org (J.G.); mehrarab@mskcc.org (B.M.); 2Boston University School of Medicine, Boston, MA 02118, USA; nsobti@bu.edu; 3Montefiore Medical Center, New York, NY 10467, USA; dnash@montefiore.org; 4Brigham and Women’s Hospital, Boston, MA 02115, USA; aklass@mcmaster.ca (A.K.); apusic@bwh.harvard.edu (A.P.)

**Keywords:** quality of life, lymphedema, lymphovenous bypass, lymph node transplant, patient-reported outcomes

## Abstract

Introduction: Analysis of quality of life (QOL) outcomes is an important aspect of lymphedema treatment since this disease can substantially impact QOL in affected individuals. There are a growing number of studies reporting patient-reported outcomes (PROMs) for patients with lymphedema. The purpose of this study was to conduct a systematic review of outcomes and utilization of PROMs following surgical treatment of lymphedema. Methods: A literature search of four databases was performed up to and including March, 2019. Studies included reported on QOL outcomes after physiologic procedures, defined as either lymphovenous bypass (LVB) or vascularized lymph node transplant (VLNT), to treat upper and/or lower extremity primary or secondary lymphedema. Results: In total, 850 studies were screened—of which, 32 studies were included in this review. Lymphovenous bypass was the surgical intervention in 16 studies, VLNT in 11 studies, and both in 5 studies. Of the 32 total studies, 16 used validated survey tools. The most commonly used PROM was the lymph quality of life measure for limb lymphedema (LYMQOL) (12 studies). In the remaining four studies, the upper limb lymphedema 27 scale (ULL27), the short form 36 questionnaire (SF-36), the lymphedema functioning, disability and health questionnaire (Lymph-ICF), and lymphedema life impact scale (LLIS) were each used once. QOL improvement following surgical treatment was noted in all studies. Conclusions: Physiologic surgical treatment of lymphedema results in improved QOL outcomes in most patients. The use of validated PROM tools is increasing but there is no current consensus on use. Future research to evaluate the psychometric properties of PROMs in lymphedema is needed to guide the development and use of lymphedema-specific tools.

## 1. Introduction

Lymphedema is a dreaded chronic disease affecting more than 5 million people in the United States [1]. Although primary lymphedema can arise from congenital or genetic mutations, the most common cause of lymphedema in Western countries is secondary to lymphatic injury in the course of surgical management of cancer (secondary lymphedema). The rates of lymphedema development following cancer treatment vary widely depending on the length of follow up and the methods used to define or measure lymphedema; however, some studies have reported lifetime rates as high as 50% following axillary lymph node dissection [2]. Further, while breast cancer is the most common cause of secondary lymphedema due to the high prevalence of this malignancy, lymphedema also occurs commonly in patients treated for other solid tumors including melanoma (16%), gynecological cancers (20%), genitourinary tumors (10%), and head/neck malignancies (4%) [3]. Risk factors can include radiation, large radiation field, conventional fractionation radiation, obesity, age, chemotherapy infusion to the affected limb, taxane-based chemotherapy, advanced stage disease, number of lymph nodes removed, and number of positive lymph nodes. [2,4,5,6,7,8,9,10,11] Secondary lymphedema can also result from traumatic injury or infections involving the lymphatic tree. Less frequently, secondary lymphedema can develop in patients due to extreme obesity. 

Regardless of etiology, patients with lymphedema experience a variety of symptoms including swelling, pain, decreased range of motion, depression and anxiety [12,13,14]. These symptoms substantially impact quality of life (QOL) and are an important clinical aspect of this disease, resulting in negative changes in functional, social, and psychological domains. Importantly, some patients have profoundly decreased QOL even without significant changes in extremity circumference [15]. These findings suggest, therefore, that assessment of QOL is an important aspect of any study aiming to analyze outcomes following surgical treatment of lymphedema. 

There are a number of patient-reported outcome measures (PROMs) available to study QOL in patients with lymphedema. Some of these instruments are lymphedema specific while others are more generic. Lymphedema-specific tools include the lymph quality of life measure for limb lymphedema (LYMQOL) [16], the upper limb lymphedema 27 scale (ULL27) [17], the lymphedema functioning, disability and health questionnaire (Lymph-ICF) [18], and the lymphedema life impact scale (LLIS) [19]. The short form 36 questionnaire (SF-36), is well known and widely used, but not specific to lymphedema [20]. Each tool is distinctive in its examination of QOL in patients with lymphedema and there is currently no consensus on which instrument to use for surgical patients.

While there is no cure for lymphedema, recent surgical treatments aiming to improve lymphatic drainage have gained popularity. These so called physiologic procedures include lymphovenous bypass (LVB), in which lymphatic channels are anastomosed to nearby veins to bypass zones of obstruction, and vascularized lymph node transplant (VLNT), in which lymph nodes are transplanted along with their blood supply to the lymphedematous limb. Interestingly, the majority of studies reporting on these procedures suggest that the best outcomes are obtained in patients with early stage disease and limited limb swelling [21]. This makes intuitive sense since it is widely accepted that early intervention for most diseases is associated with better outcomes. However, this fact also presents a clinical challenge in measuring outcomes since patients with early stage disease tend to have relatively small increases in limb volume excess. Thus, objective outcomes focusing on improvements in limb swelling may not fully capture the positive benefits of surgical intervention as reflected by changes in QOL. This problem is accentuated by the fact that there is no correlation between limb volume excess and impairments in QOL. As a result, these issues underline the importance of PROMs in the assessment of outcomes following lymphatic reconstruction. Nevertheless, to our knowledge, there has been no systematic review focusing on QOL after surgical treatment for lymphedema. Therefore, the purpose of this study was to perform a systematic review of subjective outcomes following LVB or VLNT and to analyze trends in PROM use in the literature.

## 2. Methods

A systematic review of contemporary peer-reviewed literature was performed to evaluate the QOL outcomes in the physiologic surgical treatment of lymphedema. On March 7, 2019, four databases were searched: Medline (PubMed), Embase.com, the Cochrane Library (Wiley), and Health and Psychosocial Instruments (Ovid). In all databases but Health and Psychosocial Instruments, the search had two main categories, combined using the AND operator: (1) lymphedema and (2) lymphovenous anastomosis or lymph node transplant. The search in Health and Psychosocial Instruments looked only for lymphedema-related instruments. In PubMed and Embase, we used the *Cochrane Handbook* filter for excluding animal-only studies [22]. We saved all references to the citation management software EndNote and removed duplicates following the Bramer Method [23].

Two reviewers independently reviewed 850 abstracts after removal of duplicates and 105 full texts. Clinical studies describing QOL outcomes after surgical treatment of extremity lymphedema with either LVB or VLNT, with a minimum sample size of four patients and written in English were included in our study. Non-referenced articles, case reports, review articles and non-human articles were excluded. A total of 32 studies matched inclusion criteria (Figure 1). 

Data extracted from each study included number of patients, etiology of lymphedema, stage of lymphedema, upper versus lower extremity, type of surgical procedure, other therapies used, donor site for VLNT, and follow up time/tool used for QOL assessment. Descriptive and summary statistics were used to evaluate the articles. Pearson correlation coefficient was used to assess the use of validated and ad-hoc survey tools over time.

## 3. Results

QOL was reported as an outcome measure after physiologic surgical treatment for lymphedema in 32 articles involving 954 patients. Weighted average follow-up time was 9.2 months. LVB was the primary surgical treatment in 18 studies, and VLNT in 14 studies. All studies showed an improvement in QOL (range 50–100%). Individual patient data was reported in 18 studies, totaling 717 patients. Between 50% and 100% of patients showed improvement. One-half of the studies we reviewed (n = 16) used a QOL tool without evidence of a development or psychometric validation process (i.e., ad-hoc instrument). PROMs were used in the remainder of studies and included LYMQOL (n = 12, 38%), the ULL27 (n = 1, 3%), the Lymph-ICF (n = 1, 3%), the LLIS (n = 1, 3%) and the SF-36 (n = 1, 3%; (Table 1). Over time, the proportion of studies utilizing validated tools increased (r = 0.5), while the proportion of studies using an ad-hoc questionnaire decreased (r = -0.5) (Figure 2). 

### 3.1. Patient-Reported Outcomes after LVB

#### 3.1.1. Ad-Hoc Patient Questionnaires

Twelve studies analyzing outcomes following LVB assessed QOL outcomes with ad-hoc questionnaires (Table 2). O’Brien et al. analyzed outcomes of 46 upper extremity and 6 lower extremity LVB procedures [24]. At an average follow up of 4.2 months, 38 of 52 (78%) patients experienced subjective improvement including a decrease in size, better fitting clothes, softer skin and decreased frequency of cellulitis. A few patients, 3 of 52 (6%) felt they were worse. Most patients, 83%, were able to discontinue conservative measures post-operatively. Demirtas et al. performed LVB in the lower extremities of 42 patients [25]. At an average follow up of 11.8 months, 40 of 42 (95%) patients were satisfied with the result and felt improved in terms of decreased size, decreased weight of the limb, softer, better texture of skin, easier fitting of clothes and decreased infections. Auba et al. performed LVB of either the upper or lower extremity in 10 patients [26]. Qualitative evaluation was done by recording subjective symptoms that patients reported during follow up. At 18 months follow up, 9 of 10 (90%) patients reported noticeable improvements in their symptoms (skin induration, sensation of swelling, worsening in the summer, requirement of garments, difficulty wearing clothing, numbness, erythema and mobility). Ayestaray et al. studied 20 patients with lymphedema of the upper extremity [27]. LVB was performed and at 6 months post-operative, 19 of 20 (95%) patients had improvements in their soft tissues. They also noted that 18 of 20 (90%) patients moved to a better QOL, although there is no mention as to how QOL was measured. Chang et al. reported that 19 of 20 patients (95%) that underwent LVB of the upper extremity had improvements in the symptoms of lymphedema (arm was lighter, softer and less painful) shortly following surgery [28]. However, at 12 months follow up, these findings were sustained in only 16 of 20 (80%). In another study, Chang et al. reported on outcomes in 100 consecutive patients treated with LVB for either upper or lower extremity lymphedema [29]. Average follow up for upper extremity patients was 30.4 months and 96% of patients reported their arm felt lighter, softer, and less painful. Average follow up for lower extremity patients was 18.2 months and 57% noted symptomatic improvement. Poumellec et al. reported that at 12.7 months after LVB for upper extremity lymphedema, substantial functional improvement was noted in 17 of 31 (55%) patients, and moderate improvement in 9 (29%) patients [30]. 

Three studies described creation of ad-hoc study specific tools. Chung et al. created their own questionnaire in a retrospective study of 18 patients undergoing LVB for either upper or lower extremity lymphedema [31]. The questionnaire had eight questions, scored from 0 to 5, with higher scores being better than lower scores. Three questions were related to volume, two questions were related to softness, and three questions were related to overall satisfaction. The questionnaire was administered at 6 months post-operatively. Scores were compared between patients with upper versus lower extremity lymphedema and between Campisi stage 2 and 3/4. Generally, patients with stage 2 upper extremity lymphedema had the highest average scores. Patients with stage 3/4 lower extremity lymphedema had the lowest average scores. Mihara et al. also created a study specific tool evaluating sensations of pain, strange feelings, and tension [32]. In 6 patients with secondary lower extremity lymphedema, all patients (100%) noted improvements at 6 months post-operatively. In another study by Mihara *et al,* a study specific tool was used to inquire about limb softness, pain, and severity of lower extremity lymphedema [33]. At 18.3 months, 67 of 84 (80%) patients had improvement and 4 felt worse. 

Two studies reported outcomes of LVB, with or without VLNT, using ad-hoc study specific tools. Chen et al. used a study specific tool evaluating severity of lymphedema symptoms and degree of disability and reported that these symptoms are reversed by LVB in 19 patients and VLNT from the groin (1 patient) or supraclavicular region (2 patients) at 12 months follow up [34]. Significant improvement in scores was noted from pre-operative to post-operative (*p* < 0.01). Masia et al. used a study specific tool evaluating episodes of lymphangitis, pain, swelling, heaviness, loss of sensitivity, loss of mobility, anxiety/depression, impact on daily activities, and the use of conservative therapies [35]. Two hundred patients with upper extremity lymphedema were included in the study having the following procedures: LVB (81 patients), VLNT from the groin (7 patients), DIEP/SIEA with groin lymph nodes (16 patients), LVB + VLNT (44 patients), liposuction (52 patients). Of all patients, 192 (96%) reported subjective improvement, 8 patients reported no change.

#### 3.1.2. LYMQOL

Four papers reported on LVB outcomes using the LYMQOL (Table 3). Winters *et al*, in two separate studies—one with 29 patients and 12 month follow up and another with 12 patients and 6 month follow up—compared pre-operative to post-operative LYMQOL surveys in patients with upper extremity lymphedema [36,37]. In both studies, all subscales and the overall quality of life improved significantly (*p* < 0.01). Gentileschi et al. reported a significant increase in the average overall score using the LYMQOL at 6 months after LVB for treatment of upper extremity lymphedema in 16 patients (*p* < 0.001) [38]. Salgarello et al. used the LYMQOL to evaluate outcomes of LVB for treatments of either upper extremity (n = 44) or lower extremity (n = 26) lymphedema at an average of 8.5 months [39]. Significant improvements were noted in the overall score as well as all subscales (*p* < 0.01). 

#### 3.1.3. Lymph-ICF

Cornelissen et al. performed LVB on 20 patients with upper extremity lymphedema and the Lymph-ICF was used to evaluate outcomes at 12 months [40]. Significant improvements in all subscales and total score was seen (*p* < 0.05). (Table 3).

#### 3.1.4. SF-36

Damstra et al. performed LVB on 10 patients with upper extremity lymphedema [41]. SF-36 was completed pre-operatively and at 6 months post-operatively. At a follow up of 6 months, 5 of 10 patients (50%) felt less disabled on the SF-36 questionnaire. (Table 3).

### 3.2. Patient-Reported Outcomes after VLNT

#### 3.2.1. Ad-Hoc Patient Questionnaires 

Four studies analyzing VLNT assessed QOL outcomes with no specific validated tool (Table 4). Gharb et al. evaluated 21 patients with upper extremity lymphedema who underwent VLNT using groin lymph nodes [42]. Ten patients also had liposuction. At an average of 43.1 months, average scores on a visual analog scale improved, although not significantly. Nguyen et al. report their series of 42 patients who had vascularized omentum flaps for treatment of either upper or lower extremity lymphedema, with 55% of patients also having LVB [43]. Average follow up was 14 months and they report subjective improvements in swelling, fatigue, heaviness, tightness, stiffness, sleep loss, aching, and skin quality in 35 of 42 patients (83%). There is no mention as to how data regarding symptoms was obtained and when.

Coriddi et al. analyzed results of the vascularized jejunal mesenteric lymph node transplant in 15 patients with either upper or lower extremity lymphedema [44]. One patient suffered a flap loss. At a mean follow up of 9.1 months, 12 of 14 patients had subjective improvement (85.7%). Further details are not mentioned. Dionyssiou et al. published their results of a randomized control trial comparing VLNT from the groin to the upper extremity compared to conservative treatment over a follow up period of 12 months [45]. A visual analogue scaling system (1–10) was used to assess pain, heaviness and functional disturbances. All 18 patients (100%) who received VLNT reported significant improvement at 12 months from pre-operative measurements in the pain, heaviness and function scales (*p* < 0.001). Additionally, when comparing groups, the scores for pain, heaviness and function were significantly better at 12 months in the VLNT group compared to the conservative management group (*p* < 0.001). 

#### 3.2.2. LYMQOL 

Eight papers reporting on VLNT utilized the LYMQOL in outcome analysis (Table 5). Ciudad et al. report their results of transplant of the right gastroepiploic lymph node flap to either the groin (5 patients) or axilla (5 patients) [46]. When comparing average pre-operative scores with average one year post-operative scores, significant improvements in all subscales as well as the overall score were noted (p<0.01). Patel et al. examined 25 patients, 15 upper extremity and 10 lower extremity, with either primary or secondary lymphedema [47]. VLNT was done using either a groin or submental lymph node flap. Average LYMQOL scores were recorded pre-operatively and at 1, 3, 6, 9 and 12 months post-operation. In both the upper and lower extremity patients, all subscales were improved and reached statistical significance by 9 months post-operation (range p<0.01 to p<0.05). Asuncion et al. use LYMQOL pre-operatively and at 12 months post-operatively to evaluate outcomes of 15 patients who underwent submental VLNT to either upper or lower extremity [48]. At 12 months, there was a significant improvement in the average score for all subscales and overall QOL (range *p* < 0.02 – p < 0.04). Ciudad et al. reported their results following double gastroepiploic VLNT combined with soft tissue de-bulking [49]. At 12 months post-operation, significant improvements in the average scores in all subscales as well as the overall score were noted for both the upper and lower extremity groups (*p* < 0.01). Visconti et al. reported the technique and outcomes of compartmental dual lymph node transplant from the right supraclavicular area to the lower extremity in 10 patients with lymphedema [50]. LYMQOL surveys were completed pre-operatively and at 6 and 12 months post-operatively. All patients reported improvement in QOL. Average scores for all subscales and overall score improved from the pre-operative survey to the 6 month post-operative survey, and again from the 6 month survey to 12 month survey. Maruccia performed VLNT using groin or gastroepiploic lymph nodes to the upper extremity in 39 patients [51]. In 18 of the 39, scar release and fat grafting were performed as well. At 12 months follow up, LYMQOL average scores for both groups were significantly improved (*p* < 0.001). 

Gratzon et al. used the LYMQOL and also two paper specific scales (pain and heaviness) to evaluate outcomes in 50 patients [52]. The pain scale and heaviness scales were both based on a standard 1–10 ratings. Evaluation was done pre-operatively and at 1, 3, 6, 9 and 12 months post-operatively. Twenty-four patients completed follow up at one year and had either a groin, lateral thoracic or supraclavicular VLNT to the upper extremity for secondary lymphedema. In terms of LYMQOL outcomes, all subscales and the overall score reached significant improvement by 3 months post-operation (*p* < 0.01). Using the paper specific pain and heaviness scales, pain reached significant (*p* < 0.01) improvement at 1 month post-operation and heaviness reached significant (*p* < 0.01) improvement at 1 month post-operation.

Cheng et al. evaluated 19 patients using the LYMQOL—15 who underwent submental VLNT and 4 who underwent LVB [53]. At 12 months follow up, average scores for all domains and the overall QOL score improved significantly in the VLNT groups (*p* < 0.05). The LVB group showed improvement in the overall score, and the function, appearance, symptoms and mood domains (*p* = 0.07, 0.1, 0.1, 0.07, 0.07 respectively).

#### 3.2.3. ULL27 

One study utilized the ULL27 in outcome analysis (Table 5). De Brucker et al. performed VLNT to the axilla of the affected arm using the groin as the flap donor site [54]. At an average follow up time of 29 months (range 8–64 months), there was a statistically significant improvement in the mean overall score of the ULL27 (*p* < 0.001). Average post-operative scores for each of the ULL27 domains (physical, psychological, social) also improved. Looking at individual data, 21 of 25 patients had improved scores, 3 patients had no change, and 1 patient was worse. 

#### 3.2.4. LLIS

The LLIS was used to assess outcomes in 14 patients who underwent VLNT using a muscle-sparing latissimus dorsi flap [55]. Inbal et al. report at a mean follow up of 6.7 months (range 3–12 months), 10 patients (91%) reported symptomatic improvement including softer, lighter and less painful extremities than prior to surgery. Average LLIS scores improved over time. However, of 14 patients, only 9 patients presented for evaluation at 3 months follow up and only 5 patients at 12 months. (Table 5).

## 4. Discussion

QOL improvement was reported in all papers evaluated in this systematic review examining outcomes after LVB and/or VLNT in patients with either upper or lower extremity secondary or primary lymphedema. QOL is an important outcome measure in lymphedema surgery. Previous studies have shown that although volume changes may not be evident, patients with lymphedema can have significant QOL impairments. Additionally, the severity of volume change does not necessarily correlate with subjective outcomes, as patients with relatively minor volume changes can experience very significant psychosocial and/or physical challenges [15,56]. Hormes et al. found in his study of 295 women, arm swelling and lymphedema severity were less correlated with QOL than total number of arm symptoms and specific individual symptoms [15]. A study of breast cancer patients treated with neoadjuvant chemotherapy and axillary lymph node dissections showed slight agreement, with the kappa coefficient ranging from 0.05 to 0.09, when examining lymphedema symptoms of arm swelling and heaviness compared to volume or circumference measurements [57]. At our institution, we have seen some patients with minimal or low excess volumes measurements report a high degree of impairment in QOL [56]. Despite these facts, the vast majority of studies reporting outcomes following surgical treatment of lymphedema have focused primarily on objective changes in limb volume or fluid content while a relative minority have also reported on PROMs. To our knowledge, this is the first systematic review of QOL changes following surgical physiologic lymphatic reconstruction.

Our systematic review found that QOL improvements can occur relatively early following surgery. The average weighted follow up for all studies was 9 months. Two papers that evaluated QOL using the LYMQOL found significant improvements in as little as 1 month post-operatively. Gratzon et al. reporting outcomes following VLNT to the upper extremity, found that the subscale of symptoms on the LYMQOL is significantly improved as early as 1 month after surgery. Additionally, at one month post-operatively, pain and heaviness improved significantly when measured on their paper specific scale [52]. Patel et al. reported significant improvement on the LYMQOL function subscale 1 month following VLNT in 15 patients with upper extremity lymphedema [47]. Studies with longer term follow up are needed to ensure these findings persist and are not simply related to post-operative behavior modifications (e.g., elevation, changes in activity, or improvements in compliance with compression). For example, Chang et al. used a non-validated survey tool and reported initial improvement after LVB in 95% of patients [28]. However, improvement was only sustained in 80% at one year after surgery. 

In the eighteen papers that report individual patient data, we noted that the majority of patients (50%–100%) improve following LVB or VLNT. In four studies, all patients reported improvement. The remaining studies have a small percentage of patients who either experienced no change or were worse. This is an important finding to consider. While most studies focus on the improvements after physiologic surgical treatment of lymphedema, negative results are also important and have a direct impact on patient counseling. However, due to the progressive nature of the disease, determining the etiology of worsening QOL post-operatively is difficult. While it is possible that surgery could damage functional lymphatics leading to increased lymphedema and decreased QOL, it is also possible that these surgical interventions failed to improve lymphatic drainage and the disease, as expected, worsened over time. 

Our study shows that there is little consensus in the literature on the use of PROMs to study outcomes following lymphatic surgery. Indeed, many studies reviewed in our paper (half of the papers that met inclusion criteria), failed to use validated instruments and relied on questionnaires developed in an ad hoc manner. Validated PROMs are developed using rigorous qualitative and quantitative psychometric methods to ensure that reliable, accurate, interpretable data is obtained and that the measurement tool is well targeted to the population being studied [58]. Therefore, to examine changes in QOL specific to lymphedema, the use of a validated PROM is critical to surgical outcomes research and advancement of this field. In this study, we noted that the proportion of published studies using a validated PROM has steadily increased over time. With the exception of one prior study, physiologic surgical treatment for lymphedema research began including QOL as an outcome measure in 2009. The earlier studies more often used non-validated tools or generic tools like the SF-36, even though the ULL27 was validated in 2002. The LYMQOL was validated in 2010 and has become the most widely applied tool since. While the use of validated tools is becoming more common, some recent studies continue to use non-validated measures. Considering the number of validated tools available and the ease of accessibility, the use of a validated tool in lymphedema research should be standard. 

Among the 16 studies with validated PROMs, a variety of measures were used, including the LYMQOL (n = 12), ULL27 (n = 1), LLIS (n = 1), Lymph-ICF (n = 1), and SF-36 (n = 1). This heterogeneity makes comparison of outcomes across studies difficult. Consensus among lymphedema researchers regarding which QOL(s) to use is therefore important and necessary.

Which tool should be recommended? While the SF-36 is well known and widely used, a lymphedema-specific PROM which captures the particular psychologic, social and physical factors contributing to quality of life changes caused by lymphedema is likely more useful. When examining the efficacy of surgical procedures aimed at treating lymphedema, a lymphedema-specific evaluation tool ensures QOL changes are specific to lymphedema, compared to the use of a generic tool such as the SF-36 which would evaluate a general health status. The LYMQOL, ULL27, LLIS, and Lymph-ICF are all validated and comprehensive, lymphedema-specific questionnaires. In 2013, Pusic et al. evaluated lymphedema-specific PROMs and found the ULL27 had strong psychometric properties [59]. Since that time, however, the clinical characteristics of patients eligible for lymphedema surgery has shifted, with patients with less severe disease potentially benefitting. Given this, it is unclear whether the ULL27 continues to be well targeted across the entire spectrum of the patients. In terms of PROM contents, all evaluated physical and psychosocial issues that can affect patients with lymphedema. The LYMQOL also asks about worry, irritability, feeling of being tense, and depression that can plague those suffering from lymphedema. Similarly, the ULL27 asks about feeling sad, discouraged, distressed, or angry. The LLIS inquiries about feelings of depression, frustration or anger and the Lymph-ICF asks about feeling sad, frustrated and insecure about the future due to lymphedema but neither not touch on feelings of anxiety as directly. The LYMQOL, LLIS and Lymph-ICF are available for both upper and lower extremity evaluation, while the ULL27 is dedicated to upper extremity only. 

An important limitation of our review is that it was outside the scope of our study to critically appraise the development and validation process of the lymphedema-specific PROMs identified in our search. Future research is needed that employs the COSMIN (COnsensus-based Standards for the selection of health Measurement Instruments) methodology, now used in an increasing number of reviews of PROMs, to examine the development process and psychometric properties of each PROM. The COSMIN methodology provides guidance and criteria for judging a range of psychometric properties, including content validity, structural validity, internal consistency, cross-cultural validity, reliability, measurement error, criterion validity, construct validity and responsiveness. Research that appraises PROMs using the COSMIN approach, could help to answer the question on which is the best PROM to use in future research [60]. 

This study has additional limitations. Summarization of outcomes specific to primary versus secondary lymphedema, upper extremity versus lower extremity is difficult as results regarding quality of life are often reported grouped together, including all patients in the study. Similarly, assessment of VLNT versus LVB is difficult, as some authors perform both treatments simultaneously or include a physiologic surgery with a reductive surgery. Additionally, this review is specific to physiologic procedures to treat lymphedema. Other survey tools may be used more predominantly in other fields, such a lymphedema therapy. Also, while we do see an improvement in quality of life in this study, no studies were blinded and therefore there may be a component of placebo effect. Further, more rigorous studies using a validated PROMs, preferably one common tool across all lymphedema research, with longer follow up, are needed to confirm the improvement in QOL in patients undergoing physiologic procedures for treatment of lymphedema. 

## 5. Conclusions

Patients who suffer from lymphedema and undergo treatment with physiologic procedures such as LVB or VLNT have significant improvements in QOL. The use of lymphedema-specific PROMs has increased progressively over the past decade. This is important since QOL changes related to lymphedema may not correlate with limb volumes and because surgical treatments have the highest efficacy in patient with early stage (i.e., low volume) disease. Additional studies are needed using validated tools to confirm QOL improvements after LVB or VLNT and to guide patient selection, shared surgical decision making and future innovation.

## Figures and Tables

**Figure 1 cancers-12-00565-f001:**
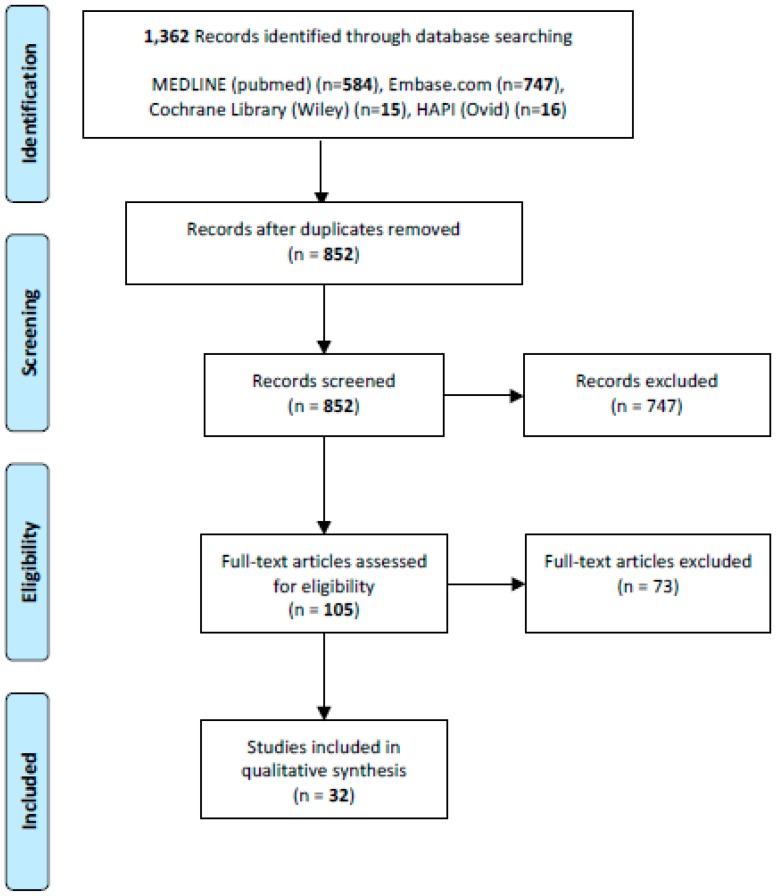
PRISMA (Preferred Reporting Items for Systematic Reviews and Meta-Analyses) chart.

**Figure 2 cancers-12-00565-f002:**
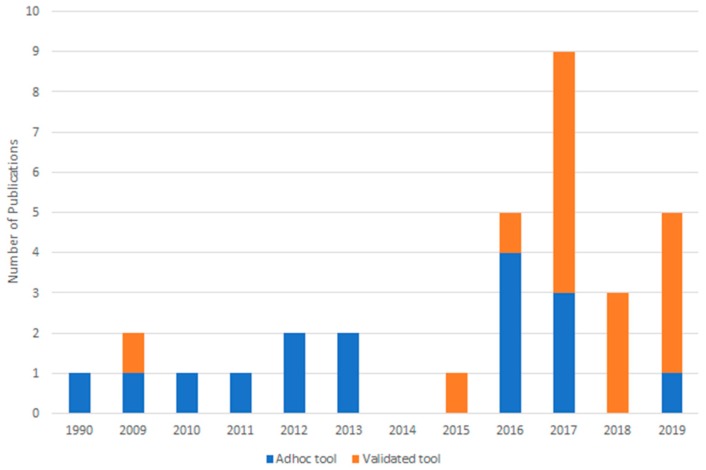
Publications with ad-hoc versus validated tools over time.

**Table 1 cancers-12-00565-t001:** Validated quality of life tools.

Validated Tool	Lymphedema Specific	Categories	Number of Questions	Lookback Period	Upper/Lower Extremity	Score Calculation
LYMQOL	Yes	Four subscales: pain, mood, function and appearance, and an additional question on overall quality of life	24 (upper) 25 (lower)	1 week (mood only)	Both	A 4-point Likert scale with additional questions that are free response. Each dimension is scored, resulting in one number for each section. The overall quality of life score is on a 1-10 scale.
ULL27	Yes	Three subscales: physical, psychological, and social	27	4 weeks	Upper	A 5-point Likert scale. Each dimension is scored, resulting in one number for each section.
Lymph-ICF	Yes	Five subscales: physical function, mental function, household activities, mobility activities, and life and social activities	29	2 weeks	Both	An 11-point Likert scale. Each dimension is scored, resulting in one number for each section.
LLIS	Yes	Three subscales: physical, psychosocial, functional, and an additional question on infection occurrence	18	1 week	Both	A 5-point Likert scale. Each dimension is scored, resulting in one number for each section.
SF-36	No	Eight subscales: physical functioning, role limitations as a result of physical problems, bodily pain, general health perception, vitality, social functioning, role limitations due to emotional problems, and mental health	36	4 weeks	Both	The domains are combined to create a physical component score and a mental component score.

**Table 2 cancers-12-00565-t002:** Quality of life (QOL) in lymphovenous bypass (LVB) using ad-hoc tools.

Study	Year Published	Number of Patients	Stage	Lymphedema Site	Primary vs. Secondary	Surgical Procedure	Baseline QOL Measure Administered Pre-Operatively?	Average Follow-Up Time	QOL Measure	Percent of Patients with Subjective Improvement
O’Brien	1990	52	Not mentioned	UE (46), LE (6)	Secondary	LVB	Not mentioned	4.2 mo	Ad-hoc tool	73%
Demirtas et al.	2009	42	Campisi stage II (12), III (17), IV (13)	LE	Secondary (34), Primary (8)	LVB	Not mentioned	11.8 mo	Ad-hoc tool	95%
Chang	2010	20	Capisi stage II (10), III (10)	UE (20)	Secondary	LVB	Yes	12 mo	Ad-hoc tool	80%
Auba et al.	2012	10	Campisi stage II (2), III (8)	LE (4), UE (6)	Secondary (9), Primary (1)	LVB	No	18 mo	Ad-hoc tool	90%
Mihara et al.	2012	6	ISL 0 (3), 1 (3)	LE	Secondary	LVB	Yes	6 mo	Ad-hoc tool	100%
Ayestaray et al.	2013	20	Campisi stage II (9), III (7), IV (3), V (1)	UE	Not mentioned	LVB	Yes	6 mo	Ad-hoc tool	95%
Chang et al.	2013	100	ICG classification stage 1 or 2 (16), 3 or 4 (14). Not all patients classified	UE (89), LE (11)	Secondary	LVB	Not mentioned	30.4 mo (UE), 18.2 mo (LE)	Ad-hoc tool	96% (UE), 57% (LE)
Mihara et al.	2016	84	ISL 1 (30), 2a (39), 2b (36), 3 (23)	LE	Primary (15), Secondary (69)	LVB	Yes	18.3 mo	Ad-hoc tool	80%
Chen et al.	2016	21	Campisi I and II (9), III (4), IV (8)	UE (13), LE (8)	Primary (4), Secondary (17)	LVB (18) or VLNT (3)	Yes	12 mo	Ad-hoc tool	100%
Masia et al.	2016	200	Not mentioned	UE (200)	Secondary (200)	LVB (81), VLNT (7), DIEP/SIEA with groin lymph nodes (16), LVB+VLNT (44), liposuction (52)	Yes	12 mo	Ad-hoc tool	96%
Poumellec et al.	2017	31	Campisi stage 2 (18), 3 (10), 4 (3)	UE	Secondary	LVB	Yes	12.7 mo	Ad-hoc tool	84%
Chung et al.	2019	18	Campisi stage II (7), III or IV (11)	UE (8), LE (10)	Secondary	LVB	No	6 mo	Ad-hoc tool	Individual patient data not reported

ISL, international society of lymphology; UE, upper extremity; LE, lower extremity; LVB, lymphovenous bypass; VLNT, vascularized lymph node transplant; DIEP, deep inferior epigastric perforator flap; SIEA, superficial inferior epigastric perforator flap; mo, months.

**Table 3 cancers-12-00565-t003:** QOL in LVB using validated tools.

Study	Year Published	Number of Patients	Stage	Lymphedema Site	Primary vs. Secondary	Surgical Procedure	Baseline QOL Measure Administered Pre-Operatively?	Average Follow-Up Time Regarding Subjective Assessment	QoL Measure	Percent of Patients with Subjective Improvement
Damstra et al.	2009	10	Campisi stage III (10)	UE	Secondary	LVB	Yes	6 mo	SF-36	50%
Cornelissen et al.	2017	20	ISL 1 (1), 2a (19)	UE	Secondary	LVB	Yes	12 mo	Lymph-ICF	Individual patient data not reported
Gentileschi et al.	2017	16	ISL 2a (7), 2b (9)	UE	Secondary	LVB	Yes	6 mo	LYMQOL	Individual patient data not reported
Winters et al.	2017	29	Campisi stages 1b–2a	UE	Secondary	LVB	Yes	12 mo	LYMQOL	Individual patient data not reported
Salgarello et al.	2018	74	Not mentioned	UE (44), LE (26)	Primary (5), Secondary (55)	LVB	Yes	8.5 mo	LYMQOL	Individual patient data not reported
Winters et al.	2019	12	Campisi stages 1–2a	UE	Secondary	LVB	Yes	6 mo	LYMQOL	Individual patient data not reported

ISL, international society of lymphology; UE, upper extremity; LE, lower extremity; LVB, lymphovenous bypass; mo, months; SF-36, short form 36 questionnaire; Lymph-ICF, lymphedema functioning, disability and health questionnaire; LYMQOL, lymph quality of life measure for limb lymphedema.

**Table 4 cancers-12-00565-t004:** QOL in VLNT using ad-hoc tools.

Study	Year Published	Number of Patients	Stage	Lymphedema Site	Primary vs. Secondary	Surgical Procedure	Donor Site (Lymph Node Transplant)	Baseline QOL Measure Administered Pre-Operatively?	Average Follow-Up Time Regarding Subjective Assessment	QOL Measure	Percent of Patients with Subjective Improvement
Gharb et al.	2011	21	Not mentioned	UE	Secondary	VLNT (10 also liposuction)	Groin	Yes	43.1 mo	Ad-hoc tool	Individual patient data not reported
Dionyssiou et al.	2016	18	ISL stage II (18)	UE	Secondary	VLNT	Groin	Yes	12 mo	Ad-hoc tool	100%
Coriddi et al.	2017	15	Not mentioned	UE (8), LE (7)	Secondary	VLNT	Jejunal Mesentery	Not mentioned	9.1 mo	Ad-hoc tool	86%
Nguyen et al.	2017	42	Modified ICG stage 3 (9), 4 (18), 5 (15)	UE (19), LE (24)	Secondary (37), Primary (2), not mentioned (3)	VLNT (55% also having LVA)	Omentum	Not mentioned	14 mo	Ad-hoc tool	83%

ISL, international society of lymphology; UE, upper extremity; LE, lower extremity; LVB, lymphovenous bypass; VLNT, vascularized lymph node transplant; mo, months.

**Table 5 cancers-12-00565-t005:** QOL in VLNT using validated tools.

Study	Year Published	Number of Patients	Stage	Lymphedema Site	Primary vs. Secondary	Surgical Procedure	Donor Site (Lymph Node Transplant)	Baseline QOL Measure Administered Pre-Operatively?	Average Follow-Up Time Regarding Subjective Assessment	QoL Measure	Percent of Patients with Subjective Improvement
Patel et al.	2015	25	Scale specific to this paper, I (1), II (6), III (13), IV (5)	UE (15), LE (10)	Secondary (23), Primary (2)	VLNT	Groin or submental	Yes	12 mo	LYMQOL	Individual patient data not reported
De Brucker et al.	2016	25	Stage 1 and 2 (classification system and number of patients per stage not mentioned)	UE	Secondary	VLNT	Groin	No	29 mo	ULL27	84%
Ciudad et al.	2017	10	ISL stage II (2), ISL stage III (8)	UE (5), LE (5)	Secondary	VLNT	Right gastroepiploic lymph node flap	Yes	14.7 mo	LYMQOL	Individual patient data not reported
Gratzon et al.	2017	24	Not mentioned	UE	Secondary	VLNT	Groin, lateral thoracic, supraclavicular	Yes	12 mo	LYMQOL and study specific pain scale and heaviness scale with standard 1-10 rating	Individual patient data not reported
Inbal et al.	2017	11	ISL stage 1 (4), 2 (7)	UE (8), LE (3)	Secondary	VLNT (64% also having LVB)	Thoracic on muscle sparing latissimus dorsi flap	Yes	6.7 mo	LLIS	91%
Cheng et al.	2018	19	Not mentioned	LE (19)	Primary (19)	VLNT (15), LVB (4)	Submental	Yes	12 mo	LYMQOL	Individual patient data not reported
Asuncion et al.	2018	15	Not mentioned	LE (10), UE (4), both LE/UE (1)	Not mentioned	VLNT	Submental	Yes	12 mo	LYMQOL	Individual pre-operative patient data not reported
Ciudad et al.	2019	16	ISL stage III (16)	UE (6), LE (10)	Secondary (15), Primary (1)	VLNT (with debulking)	Gastroepiploic	Yes	12 mo	LYMQOL	Individual patient data not reported
Maruccia et al.	2019	39	Not mentioned	UE	Secondary	VLNT (18 also with axillary scar release and fat grafting)	Groin (20), Gastroepiploic (19)	Yes	12 mo	LYMQOL	Individual patient data not reported
Visconti et al.	2019	10	ISL 2b (10)	LE	Secondary	VLNT	Supraclavicular	Yes	12 mo	LYMQOL	100%

ISL, international society of lymphology; UE, upper extremity; LE, lower extremity; LVB, lymphovenous bypass; VLNT, vascularized lymph node transplant; mo, months.

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
