# Peer review of "Systematic Review of Patient-Reported Outcomes following Surgical Treatment of Lymphedema"

_cancers, 2020, doi:10.3390/cancers12030565_

Round 1

Reviewer 1 Report

This is a nice and comprehensive study about one of the major challenges in the clinical management of cancer survivors.

I have the following comments

-Figure 1 is unreadable because of text overlaps

-The Tables have a lot of abbreviations that are only explained in the text, making extremely difficult to understand the tables. To allow for a clear reading, please state all the abbreviations in legends

-In the Introduction section only surgical explenations of lymphedema are provided, please expand with additional risk factors (laterality, tumor-specific pathological features, taxanes, trastuzumab, ...)

Author Response

Thank you for your comments.

Regarding Figure 1, a word document version was provided so formatting can be adjusted.

Regarding tables, legends were added.

Regarding additional risk factors for lymphedema, these were added to the introduction.

Reviewer 2 Report

Dear authors,

Thank you for this comprehensive systematic review of PROMs following surgical treatment. This is a much needed area of further research and consensus on using validated QOL tools to assess surgical treatment outcomes.

1) I think it would be very helpful if a column could be added in Table 2 that will specify if the tool was administered before / pre-surgery. At the moment I can only see two references in the text that specify if a tool was administered pre-surgery. This is very important in measuring outcome of the surgical intervention next to the follow up time which you've reported on.

2) I am wondering why you have not included liposuction which is a widely accepted surgical procedure for lymphedema? There are two studies in your review that also have liposuction (next to other surgical procedure) and you then report the outcome of their whole cohort rather than focus on the LVB and VLNT. This is a bit confusing - maybe exclude those studies with liposuction as a treatment or mention the outcome of your targeted cohort only (e.g LVB -VLNT) or include all liposuction studies.. Please consider the same for the 'scar release and fat grafting outcome' (ref 42 Maruccia -line 232/233).

3) Line 14, abstract, please specify PROM abbreviation  

4) Line 336 , typo - 'lyphedema'

Author Response

Thank you for your comments

-We agree that specific notation of surveys administers pre-operatively is important for this study. Therefore, we have added a column to tables 2, 3, 4 and 5 stating if the QOL tool was administered pre-operatively.

-This paper was aiming to focus on physiologic treatments of lymphedema and not treatments aimed at debulking (liposuction, excision). As liposuction/debulking can influence results, we ensured that if these procedures were done as well, this was noted in both the tables and in the manuscript text so readers will be aware. Unfortunately, there is no way to break up the results of just those patients in most cases and we felt it was important to still include those papers in our review.

-PROM abbreviation specified

-lymphedema spelling error corrected

Round 2

Reviewer 1 Report

I acknowledge that there has not been an attempt to fix Figure 1.

However, the comment on the additional risk factors to be added to the introduction should be addressed more carefully. It is unacceptable to read a general sentence referring to two 10-year-old reviews. I believe that this manuscript is of value. Please improve the introduction with more recent information on lymphedema risk factors.

Author Response

Thank you for your comments. 

Figure 1 appears to be formatted correctly without overlapping texts when I open both the pdf and doc versions on both my computer and phone. A document that can be edited has been provided to the editor previously, should Figure 1 not be aligned properly during final formatting. 

Risks of lymphedema were expanded and 7 additional, more recent references were added. 

Thank you